# The Diverse Efficacy of Food-Derived Proanthocyanidins for Middle-Aged and Elderly Women

**DOI:** 10.3390/nu12123833

**Published:** 2020-12-15

**Authors:** Toru Izumi, Masakazu Terauchi

**Affiliations:** 1Department of Production and Quality Control, Kikkoman Nutricare Japan Incorporation, Nihonbashikoamicho 3-11, Chuo, Tokyo 103-0016, Japan; 2Department of Women’s Health, Tokyo Medical and Dental University, Yushima 1-5-45, Bunkyo, Tokyo 113-8510, Japan; teragyne@tmd.ac.jp

**Keywords:** proanthocyanidin, middle-aged, women, menopause, cardiovascular disease, grape seed, cranberry

## Abstract

Middle-aged and elderly women are affected by various symptoms and diseases induced by estrogen deficiency. Proanthocyanidins, widely present in many kinds of fruits and berries, have many beneficial effects, such as antioxidative, anti-inflammatory, and antimicrobial activities. We researched the effects of proanthocyanidins for middle-aged and elderly women, finding that it has been revealed in many clinical trials and cohort studies that proanthocyanidins contribute to the prevention of cardiovascular disease, hypertension, obesity, cancer, osteoporosis, and urinary tract infection, as well as the improvement of menopausal symptoms, renal function, and skin damage. Thus, proanthocyanidins can be considered one of the potent representatives of complementary alternative therapy.

## 1. Introduction

Menopause results from reduced secretion of the ovarian hormones estrogen and progesterone, which takes place as the finite store of ovarian follicles is depleted. Women in the menopausal transition and postmenopausal period are affected by vasomotor symptoms, urogenital atrophy, sexual dysfunction, somatic symptoms, cognitive difficulty, sleep disturbance, and psychological problems. Some of these effects such as vasomotor symptoms are closely associated with estrogen deficiency, but the exact mechanisms underlying the other symptoms are not fully understood [1]. Postmenopausal women are also at increased risk for cardiovascular morbidity [2] as a net effect of central obesity [3], dyslipidemia [4], hypertension [5], and diabetes [6] partly induced by estrogen deficiency.

Flavonoids are a class of polyphenolic compounds with significant human health benefits [7]. Some of the flavonoids such as the flavan-3-ols catechin and epicatechin polymerize to form tannins. Tannins are plant secondary metabolites that can be hydrolysable or condensed. The condensed tannins are also known as proanthocyanidins [8,9]. They are widely present in the plant kingdom, for example, in fruits, berries, nuts, seeds, and bark of pine trees [10,11,12]. Proanthocyanidins are oligomers or polymers of flavan-3-ols, where the monomeric units are linked mainly by C-4 to C-8 bonds, although less frequently C-4 to C-6 linkages can also be found. These types of linkages lead to the formation of the so-called B-type proanthocyanidins. A-type proanthocyanidins, on the other hand, are characterized by an additional bond between C-2 → C-7 of the basic flavan-3-ol units [13]. Proanthocyanidins are composed of different flavan-3-ol subunits, known as proanthocyanidin monomers or catechins. The most common monomeric units are the diastereomers of (epi)catechin, (epi)afzelechin, and (epi)gallocatechin (Figure 1).

In recent years, considerable attention has been paid to proanthocyanidins due to the potential beneficial effects on human health, including antioxidative, cardio-protective anti-inflammatory, and anticancer properties [13,14,15,16]. Some reviews have also been published on the effects of proanthocyanidins [9,13], however, there still seems to be no reviews on the effects for middle-aged and elderly women. This review therefore aims to highlight aspects of the effects of food-derived proanthocyanidins for middle-aged and elderly women’s health.

## 2. Search Strategy and Study Selection

A literature search of all English language studies published was performed using PubMed (http://www.ncbi.nlm.nih.gov/pubmed) with the addition of other scientific papers of relevance found in web sources or in previously published reviews. The search terms and strategy used for the study selection were (proanthocyanidins OR procyanidins OR flavan 3-ol polymers) AND (women OR female) AND clinical trials. Human studies were used as further criteria for the literature search. The search was limited to the last 20 years of publication. We conducted the literature search in the scientific databases and assessed and verified the eligibility of the studies on the basis of the title and abstract. Inclusion criteria: (i) randomized controlled trials (RCT), randomized cross-over trials, quasi-randomized controlled trial (qRCT); (ii) long-term (over 6 months) clinical trials; (iii) prospective, cohort, and case-control studies. The exclusion criteria were (i) studies performed in only male or only under 40-aged women; (ii) studies analyzing the relationship between other polyphenols (not proanthocyanidins) and the occurrence of disease; (iii) studies performed in in vitro or in animal models; (iv) studies reporting data on the disease that is not related to middle-aged and elderly women; and (v) published articles in a language different from English and with no accessible translation.

## 3. Results

Table 1 is a summary showing current evidence of studies regarding the effects of proanthocyanidins for middle-aged and elderly women. We report the detail results in each field as follows: menopausal disorders, cancer, hypertension, cardiovascular disease, obesity, osteoporosis, urinary tract infection, renal function, and skin damage. In addition, we unfortunately could not find any reports on the effects of the disease that increase in the postmenopausal period, such as osteoarthritis, dyslipidemia, cognitive impairment, overactive bladder, urinary incontinence, vulvovaginal atrophy.

### 3.1. Menopausal Disorders

In a randomized, double-blind, placebo-controlled study in Japanese women aged 40 to 60 years who received grape seed proanthocyanidins for 8 weeks, Terauchi et al. evaluated the effect for menopausal symptoms. The mean physical symptom score for the nine items in the physical health domain of the Menopausal Health-Related Quality of Life (MHR-QOL) questionnaire significantly improved in the high-dose (200 mg proanthocyanidins/day) group after 8 weeks of treatment (*p* < 0.05). The mean score for hot flashes similarly improved in the high-dose group after 8 weeks (*p* < 0.05). In the case of changes in psychological symptoms, the mean (Hospital Anxiety and Depression Scale (HADS)-Depression subscale score did not change significantly in any group, whereas the mean HADS-Anxiety subscale score improved in both low-dose (100 mg proanthocyanidins/day) and high-dose groups after 4 weeks of treatment. The change in HADS-Anxiety subscale score from baseline to 8 weeks was significantly higher in the high-dose group than in the placebo group (*p* < 0.05). The mean Athens Insomnia Scale also improved in the high-dose group after 8 weeks [17]. Pine bark extract proanthocyanidins have been evaluated for improvement of climacteric symptoms in two separate clinical studies. A relatively low daily dosage of pine bark extract proanthocyanidins were found to be especially effective for improving vasomotor and insomnia/sleep problem symptoms, which were significantly better after 4 and 12 weeks than with placebo in perimenopausal Taiwanese women (*p* < 0.05). Total Kupperman’s index for perimenopausal symptom severity score decreased significantly compared to placebo after 12 weeks of treatment (*p* < 0.05). Symptom score was also significantly better after 4 weeks of treatment with active as compared to placebo [18]. Yang et al. reported that in 155 peri-menopausal Taiwanese women, many climacteric symptoms of the Women’s Health Questionnaire significantly improved compared to start of treatment (*p* < 0.001) after 6 months as follows: somatic problems, depression, vasomotoric problems, memory/concentration, attractiveness, anxiety, sexual behavior, sleep, and menstrual problems [19]. In a large prospective female cohort study with 10-year of follow up, Chang et al. reported that in flavonoid-rich food-based analyses, the hazard ratio (HR) was 0.82 (95% confidence interval (CI): 0.74−0.91) among participants who consumed up to two servings of citrus fruit or juices per day compared with below 1 serving per week. In the participants alone, total flavonoids, polymers, and proanthocyanidin intakes showed significantly (9–12%) lower depression risk. In analyses among late-life participants (aged up to 65 years at baseline or during follow-up), for whom they could incorporate depressive symptoms into the outcome definition, the researchers found that higher intakes of all flavonoid subclasses except for flavan-3-ols were associated with significantly lower depression risk; proanthocyanidins showed the strongest associations (HR: 0.83; 95% CI: 0.90) [20].

### 3.2. Cancer

In the SU.VI.MAX prospective cohort study followed from 1994 to 2007 in France, Touvier et al. revealed that in 55 older-aged women who were non-to-low alcohol drinkers, the intake of proanthocyanidins was associated with significantly decreased breast cancer risk (*p* < 0.05) [21]. Rossi et al. analyzed data from an Italian case–control study including 454 incidents of histologically confirmed endometrial cancers and 908 hospital-based controls to examine the relationship between dietary flavonoids and endometrial cancer. Women in the highest quartile category of proanthocyanidins with trimers vs. the first three quartile categories had an odds ratio for endometrial cancer of 0.66 (95% CI = 0.48–0.89). High consumption of selected proanthocyanidins may reduce endometrial cancer risk [22]. In another study wherein participants were 34,708 postmenopausal women in the Iowa Women’s Health Study who completed a food frequency questionnaire and were followed for cancer occurrence from 1986 through 2004, after multivariable adjustment, lung cancer incidence was inversely associated with intakes of proanthocyanidins (HR = 0.75; 95% CI: 0.57–0.97, which resulted in the highest vs. lowest quintile). Among current and past smokers, those with intakes in the highest quintile for proanthocyanidins (HR 5 = 0.66; 95% CI: 0.49–0.89) had significantly lower lung cancer incidence than those in the lowest quintile [23].

### 3.3. Hypertension

In a randomized, double-blind, placebo-controlled study in Japanese women aged 40 to 60 years who received grape seed proanthocyanidins for 8 weeks, the mean systolic and diastolic blood pressure were significantly reduced in both low-dose (100 mg proanthocyanidin per day) and high-dose (200 mg proanthocyanidin per day) groups after 4 weeks of treatment. The change in diastolic blood pressure (DBP) from baseline to 8 weeks was significantly higher in the low-dose and high-dose groups than in the placebo group (*p* < 0.05) [17]. Additionally, Odai et al. showed in a randomized, double-blind study of 21 non-smoking middle-aged women (including men), the oral intake of grape seed proanthocyanidins significantly reduced the mean systolic blood pressure (SBP) and DBP by 13.1 and 6.5 mmHg, respectively, in the high-dose (400 mg proanthocyanidin per day) group after 12 weeks of intervention [24]. A cohort study reported the anti-hypertensive effect of proanthocyanidins on middle-aged and elderly women. In a prospective cohort of 40,574 disease-free French women who responded to a validated dietary questionnaire, there was a 9% lower rate of hypertension for women in the highest category of proanthocyanidin intake than for women in the lowest category of intake of proanthocyanidin (HR: 0.91 (95% CI: 0.85–0.97; *p*-trend = 0.0051)) [25].

### 3.4. Cardiovascular Disease

Odai et al. conducted a randomized, double-blind, placebo-controlled study on 6 men and 24 women aged 40–64 years old who received tablets containing either low-dose (200 mg/day) or high-dose (400 mg/day) grape seed extract proanhocyanidin, or placebo, for 12 weeks. In an ad hoc analysis of non-smoking participants (*n* = 21), stiffness parameter, distensibility, incremental elastic modulus (Einc), and pulse wave velocity (PWV) also significantly improved in the high-dose group after 12 weeks. Changes in Einc and PWV from baseline to 12 weeks were significantly greater in the high-dose group than in the placebo group (Einc, *p* = 0.023; PWV, *p* = 0.03) [24]. Zhao et al. performed a clinical trial to evaluate the effect of apple or apple extract polyphenol on a blood parameter, oxidized low-density lipoprotein/beta2-glycoprotein I complex (oxLDL-b2GPI), related to atherosclerosis in postmenopausal American women. Consumption of apples or apple polyphenols produced a statistically significant decrease in oxLDL-b2GPI, while placebo did not produce a significant change. The mean decrease was larger for the apples than for the apple polyphenols. The apple extract polyphenol and the apples both had their largest polyphenol contribution from proanthocyanidins.

Therefore, eating one apple containing mainly proanthocyanidins per day had a substantial effect on a blood parameter related to atherosclerosis [26]. In a prospective study in postmenopausal women in the Iowa Women’s Health Study who were free of cardiovascular disease (CVD) and had complete food-frequency questionnaire information at baseline, Mink et al. evaluated the association between flavonoid intake and CVD mortality. As a result, the intake of proanthocyanidins was significantly inversely associated with coronary heart disease (CHD) mortality in models after adjustment for age and energy (*p* < 0.05), and for total CVD mortality, a significant inverse association after adjustment for age and energy intake was observed for intake of proanthocyanidins (*p* < 0.01) [27]. McCullough et al. also reported a large U.S. cohort study for the association between flavonoid intake and CVD mortality among U.S. participants. In 60,289 women in the Cancer Prevention Study II Nutrition Cohort with a mean age of 69 years, proanthocyanidins consumption was associated with the significant great reduction in CVD risk for the highest intake quintile compared with the lowest quintile (*p* = 0.01) [28]. Jennings et al. showed the effect of proanthocyanidins for fat mass ratio (FMR) in a study of healthy middle-aged female twins. In monozygotic, intake-discordant twin pairs, twins with higher intakes of proanthocyanidins (*p* = 0.01) had a significantly lower FMR than that of their cotwins with within-pair differences of 3–4% [29]. Some meta-analysis for the association between the flavonoid intake and CVD or CHD risk has been reported so far [41,42,43,44,45], but the stratified analysis for middle-aged and elderly women have not yet been performed. Therefore, we failed to evaluate the efficacy of proanthocyanidins to women for those meta-analysis.

### 3.5. Obesity

Tresserra-Rimbau et al. assessed whether high intakes of some classes of polyphenols including proanthocyanidins were associated with type 2 diabetes (T2D) in a population with metabolic syndrome and how these associations depend on body mass index (BMI) and sex. This baseline cross-sectional analysis included 6633 participants from the PREDIMED-Plus trial. They found in a case of proanthocyanidins for elderly Spanish women that there were significant and linear inverse associations in comparing extreme quartiles for overweight and obesity, recognized as important risk factors for T2D [30]. In another cohort study in Korea, Kim et al. investigated the association between dietary flavonoid intake and the prevalence of obesity using body mass index (BMI), waist circumference, and percent body fat (%BF) among elderly Korean females. A higher total intake of flavonoids was associated with a lower prevalence of obesity in women, on the basis of %BF (odds ratio (95% CI) = 0.82 (0.71–0.94)), and abdominal obesity (0.81 (0.71–0.92)). The intake of proanthocyanidins (0.81 (0.71–0.92)) was inversely associated with abdominal obesity, and a higher intake of proanthocyanidins (0.85 (0.75–0.98)) was associated with a lower prevalence of obesity with respect to %BF in women [31]. Some meta-analyses for the association between flavonoid intake and T2D or overweight risk have been reported thus far [46,47], but the stratified analysis for middle-aged and elderly women have not yet been performed. Therefore, we failed to evaluate the efficacy of proanthocyanidins to women for those meta-analyses.

### 3.6. Osteoporosis

Panahande et al. reported the effects of pine bark extract procyanidin on bone remodeling in postmenopausal osteopenic women in a randomized, double-blinded, controlled clinical trial for 12 weeks. After the 12-week intervention, that is, pine bark extract procyanidin supplementation, a significant increase in bone alkaline phosphatase (BAP) and procollagen type 1 amino-terminal propeptide (P1NP) levels, and a significant decrease in C-terminal telopeptide of type I collagen (CTx1) were observed. Compared with the control group, pine bark extract procyanidin supplementation resulted in a significant increase in P1NP levels (*p* < 0.05), BAP levels (*p* < 0.01), and BAP/CTx1 ratio (*p* < 0.01), and a significant decrease in CTx1 levels (*p* < 0.01) [32]. Additionally, a large cross-sectional study examined the associations of dietary intake of total flavonoids and their subtypes with bone density in elderly Chinese women. Zhang et al. showed that after adjusting for covariates, women who consumed higher proanthocyanidins tended to have greater bone mineral density in the whole body, femoral neck, and lumbar spine (all trend *p* < 0.05). Women in the highest (vs. the lowest) quartile of proanthocyanidin intake had 0.014–0.016 g/cm^2^ greater BMD in the whole body, femoral neck, and lumbar spine [33].

### 3.7. Urinary Tract Infection

Takahashi et al. performed a randomized, placebo-controlled, double-blind study to examine the rate of relapse in Japanese patients with urinary tract infection (UTI) who suffered from multiple relapses when using cranberry juice containing 40 mg of proanthocyanidin for 24 weeks. In the group of females aged 50 years or more, there was a significant difference in the rate of relapse of UTI between groups cranberry juice and placebo (*p* < 0.05) [34]. Aside from this study, Maki et al. reported another clinical study for UTI recurrence using cranberry juice. In a randomized, double-blind, placebo-controlled, multicenter clinical trial, women with a history of a recent UTI were assigned to consume one 240 mL serving of cranberry beverage or a placebo beverage for 24 weeks. The annualized UTI incidence density was significantly reduced in the cranberry compared with the placebo group (incidence rate ratio: 0.62; 95% CI: 0.42–0.92; *p* < 0.05). The consumption of 41.1 mg cranberry proanthocyanidin in a beverage lowered the number of clinical UTI episodes in women with a recent history of UTI [35]. Vostalova et al. showed that the study tested whether whole cranberry fruit powder (proanthocyanidin content 0.56%) could prevent recurrent UTI in 182 women with two or more UTI episodes. Participants were randomized to a cranberry (*n* = 89) or a placebo group (*n* = 93) and received daily 500 mg of cranberry for 6 months. The number of UTI diagnoses was counted. The intent-to-treat analyses showed that in the cranberry group, the UTIs were significantly fewer (10.8% vs. 25.8%, *p* = 0.04, with an age-standardized 12-month UTI history (*p* < 0.05) [36].

### 3.8. Renal Function

Ivey et al. performed a cohort study to determine the association of habitual proanthocyanidin intake with renal function and the risk of clinical renal outcomes in a population of elderly Caucasian women. Compared to participants with low consumption, participants in the highest tertile of proanthocyanidin intake had a 9% lower cystatin C concentration (*p* < 0.001). High proanthocyanidin consumers were at 50% lower risk of moderate chronic kidney insufficiency, and 65% lower risk of experiencing a 5-year renal disease event (*p* < 0.05). Proanthocyanidin intake was associated with improved renal function and reduced risk of chronic kidney disease and renal disease-associated events [37].

### 3.9. Skin Damage

Yamakoshi et al. investigated the reducing effect of grape seed extract proanthocyanidin (GSEP) on chloasma for 12 middle-aged Japanese women in a one-year open design study. The first 6 months of GSEP intake improved or slightly improved chloasma in 10 of the 12 women (83%, *p* < 0.01) and following 5 months of intake improved or slightly improved chloasma in 6 of the 11 candidates (54%, *p* < 0.01). L* values also increased after GSEP intake (57.8 ± 2.5 at the start vs. 59.3 ± 2.3 at 6 months and 58.7 ± 2.5 at the end of the study). Melanin index significantly decreased after 6 months of the intake (0.025 ± 0.005 at the start vs. 0.019 ± 0.004 at 6 months; *p* < 0.01), and also decreased at the end of study (0.021 ± 0.005; *p* < 0.05) [38]. Another study on the effect of proanthocyanidin for melasma has been reported. Evangeline et al. showed in middle-aged Filipino women the oral intake of cranberry procyanidin and antioxidative vitamin A, C, and E significantly decreased the degree of pigmentation in the left malar and right malar regions (*p* < 0.0001), and the melasma area and severity index showed a significant improvement in the left malar and right malar regions (*p* < 0.001) [39]. The effect of pine bark extract proanthocyanidin (PBE) for human skin has been revealed. Furumura et al. performed a randomized clinical trial of oral supplementation with PBE for middle-aged Japanese women. In the low-dose trial, photoaging scores as assessed by dermatologists after 12 weeks of PBE were compared with those at the beginning of the study. Scores for solar lentigines, mottled pigmentation, roughness, wrinkles, and swelling showed significant improvement at 12 weeks [40].

## 4. Discussion

We show the scheme for the relationship between the effect of proanthocyanidins and disease field in Figure 2.

Menopausal disorders are caused by estrogen deficiency. Hormone replacement therapy (HRT) remains one of the most effective therapies for vasomotor symptoms that are representative of menopausal symptoms, and it could be beneficial for young and recently postmenopausal women in relation to improvements in cardiovascular health [48]. Soy isoflavones are well known as phytoestrogens and have been heavily reported with regard to the clinical efficacy for menopausal symptoms [49,50]. In this review, we have reported that two types of proanthocyanidins, derived from grape seed and pine bark, improved menopausal symptoms. Terauchi et al. proposed that the mechanism of alleviative effect for vasomotor symptoms is generally attributed to their antioxidant activities, although they do not bind to estrogen receptors, and also have considered that their hypnosedative and anxiolytic activities might partly explain the effects of proanthocyanidins acting through gamma amino butyric acid (GABA)A receptors [51]. Moreover, anxiolytic and depressive activities of proanthocyanidins have been reported, involving the central monoaminergic neurotransmitter systems [52] and inhibiting the expressions of the proinflammatory cytokines, iNOS and COX-2 in the hippocampus [53]. In a prospective cohort study, it has also been revealed that higher proanthocyanidin intakes were associated with lower depression risk, with the highest intake group taking up to 160–179 mg of proanthocyanidins per day. The amount was approximated to the result of intervention trials as mentioned above.

Regarding cancer prevention, laboratory experiments using animal models or cultured human cell lines support a potential role of polyphenols in cancer prevention through antioxidant, immunomodulatory, anti-inflammatory, anti-angiogenic, and pro-apoptotic properties [54,55]. However, the effect of polyphenol intake on disease prevention in humans is difficult to predict, partly because in vivo studies often employed doses or concentrations far beyond those achievable by human diet. The prevention of breast and endometrial cancer is of particularly high significance for elderly women. In two cohort studies, it was revealed that the intake of proanthocyanidins was associated with significantly decreased breast and endometrial cancer risk. Proanthocyanidins have antioxidant and antiangiogenesis effects and may influence signal transduction and inhibit the action of DNA topoisomerases [56,57]. Although the bioavailability of higher molecular weight proanthocyanidins is lower, they are characterized by a higher gastric stability [58] and a higher potential scavenger activity [59]. In fact, bioavailability of proanthocyanidins (in monomeric, oligomeric, and polymeric forms of flavan-3-ols) is influenced by their degree of polymerisation; monomers are readily absorbed in the small intestine, whereas oligomers and polymers need to be biotransformed by the colonic microbiota because they are resistant to acid hydrolysis in the stomach [58]. Therefore, phenolic metabolites, rather than the original high-molecular weight compounds found in foods, may be responsible for the health effects derived from proanthocyanidin consumption [60], especially those with higher degree of polymerization. In experimental studies, the microbial metabolites of proanthocyanidins still bearing a free phenolic acid showed protective effects against oxidative stress and obesity [61,62], the major risk factors for endometrial cancer. Culter et al. reported that lung cancer incidence was inversely associated with intakes of proanthocyanidins. In another animal experiment, it was revealed that administration of grape seed proanthocyanidins to athymic nude mice by oral gavage (5 days per week) markedly inhibited the growth of s.c. A549 and H1299 lung tumor xenografts, which was associated with the induction of apoptotic cell death, increased expression of Bax, reduced expression of anti-apoptotic proteins, and activation of caspase-3 in tumor xenograft cells [23]. These results suggest that proanthicyanidins may represent a potential component for lung cancer.

Evidence from many short-term trials in humans has suggested that flavonoids and, in particular, flavanol monomers and procyanidin may have a beneficial effect on blood pressure in humans [63]. In this review, we show the effect of grape seed proanthocyanidin on blood pressure in elderly Japanese women and the relation between proanthocyanidin intake and incidence of hypertension in a large prospective cohort of women. Odai et al. considered that grape seed proanthocyanidin improving blood pressure without affecting flow-mediated dilation indicates that the antioxidant effects of proanthocyanidins could regulate vascular tone, not through NO release, but by other endothelial responses, which results in blood pressure reduction; one study on hypertensive rats that supports their results showed the positive association between reactive oxygen species (ROS) level and pulse wave velocity, arterial wall thickness, and collagen deposition and the beneficial effects of antioxidants on arterial stiffness and remodeling [64], implying that the antioxidant capacity of grape seed proanthocyanidin could contribute to decreased ROS levels and improved vascular elasticity. Evidence for the potential mechanism by which proanthocyanidin would affect blood pressure could also be related to the inhibition of the angiotensin-converting enzyme [65].

Epidemiologic data suggest that dietary flavonoids may have beneficial cardiovascular effects in human populations. Several prospective studies have reported statistically significant inverse associations between total flavonoid intake or the intake of specific classes of flavonoids and cardiovascular disease (CVD) incidence or mortality [66,67]. In this review, we revealed that incremental elastic modulus (Einc) and pulse wave velocity (PWV) also significantly improved by oral intake of grape seed proanthocyanidins, and consumption apple proanthocyanidins produced a statistically significant decrease in oxidized low-density lipoprotein/beta2-glycoprotein I complex (oxLDL-b2GPI) related to atherosclerosis. These results support the fact that proanthocyanidins affect cardiovascular health. In cohort studies, the intake of proanthocyanidins has also been reported to significantly reduce CVD mortality for middle-aged and elderly women. The cardioprotective effects of proanthocyanidins have been highlighted by several studies regarding the mechanism: oxidative stress, cardiomyocytes and the endothelium, anti-inflammatory effects, metabolic effects, etc. [68]. These beneficial effects appear to be mediated by various signaling pathways and mechanisms acting either independently or synergistically.

Overweight and obesity are recognized as important risk factors for type 2 diabetes (T2D). Some cohort studies for elderly women have been investigated with regards to diabetes mellitus and obesity and have revealed that the intake of proanthocyanidins is inversely associated with abdominal obesity or overweight. Yang et al. proposed that proanthocyanidins lower hepatic glucose production by activating adenosine monophosphate(AMP)-activated protein kinase and/or insulin-signaling pathways, play a role in protecting pancreatic β cells from oxidative stress, and promote insulin secretion and β-cell survival, and the actions of proanthocyanidins on liver and pancreatic β cells could lower blood glucose and reduce metabolic and oxidative stress on the islets to enhance glucose homeostasis, which could be of long-term benefit to overall metabolic health [69]. Regarding the hypertension, cardiovascular, and obesity fields, the highest intake group consumed 168–524 mg of proanthocyanidins as a daily average amount in each cohort study reported in this review. These results may suggest that the effective intake of proanthocyanidins for these diseases may be larger than that for menopausal symptoms.

Osteopenia is an important predictor of osteoporosis as it is characterized by low bone mineral density (BMD) [70]. Because osteoporosis has also become an important factor in morbidity and mortality in elderly women, its prevention is therefore of utmost importance in this age group. Many studies have examined the associations between flavonoids and bone health [71,72,73]; however, most of them, including randomized controlled trials (RCTs), have focused primarily on the isoflavone subclass, which is mainly contained in soy foods. In this review, pine bark proanthocyanidin, not estrogenic compound, supplementation in postmenopausal osteopenic women has been shown to produce favorable effects on bone markers. Moreover, in a cohort study, dietary proanthocyanidin intake was positively associated with BMD in middle-aged and elderly women. The inhibition of receptor activator of nuclear factor-kappa B (NF-κB) ligand (RANKL)-dependent osteoclast differentiation caused by proanthocyanidins was indicated by studies in vitro [74]. Animal experimental results also suggested that proanthocyanidins can promote bone formation [75]. Proanthocyanidins may be expected for the prevention of osteoporosis instead of soy isoflavones.

A urinary tract infection (UTI) is common and increasingly difficult to treat because of the rising rates of antibiotic resistance [76,77]. Approximately 60% of women will experience up to one UTI in their lifetimes. Cranberry consumption has been evaluated as a strategy for reducing clinical UTI recurrence in women with a recent history of a UTI [78,79]. In this review, we have shown that the consumption of cranberry juice or cranberry powder containing proanthocyanidins lowered the incidence of UTI for elderly women. The proanthocyanidins in cranberry have been reported to inhibit the growth of several pathogenic bacteria, such as uropathogenic *Escherichia coli*, cariogenic *Streptococcus mutans*, and oxacillin-resistant *Staphylococcus aureus* [80]. The cranberry proanthocyanidins, consisting primarily of epicatechin tetramers and pentamers with at least one A-type linkage, have been found to be active against the pathogenic bacteria. In addition, the daily intake of 65% cranberry juice is recommended to prevent recurrent cystitis for postmenopausal women in the Guideline in The Japanese Association for Infectious Disease/Japanese Society of Chemotherapy (JAID/JSC) Guidelines for Infection Treatment—Urinary Tract Infection.

Chronic kidney disease (CKD) represents a growing public health issue [81]. Oxidative stress, atherogenesis, nitric oxide homeostasis, and endothelial function play important roles in the pathogenesis of this disease [82,83,84,85,86]. Ageing is associated with structural and functional changes in the kidneys [87], resulting in impaired renal function [88]. Cystatin C provides early indications of renal dysfunction [89]. Ivey et al. indicated in a cohort study that higher intake of proanthocyanidins lowered the plasma cystatin C level. There is direct evidence that proanthocyanidins can specifically improve renal health in animal models by reducing oxidative stress, improving antioxidant defense potential, and reducing oxidative renal injury [56,90,91]. Proanthocyanidins may contribute to the prevention of severe CKD.

Abnormal facial pigmentation such as chloasma (melasma) is often of great cosmetic importance to women. Chloasma is a common acquired symmetrical hypermelanosis characterized by irregular light to dark brown macules and patches on sun-exposed areas of the skin. Although the etiology is unknown, several etiogenic factors have been implicated, including genetic factors, UV exposure, pregnancy, hormonal therapies, cosmetics, phototoxic drugs, and antiseizure medications [92]. In this review, we have reported that grape seed proanthocyanidins and cranberry proanthocyanidin with vitamin A, C, and E improved chloasma in an intervention study. Yamakoshi et al. considered that grape seed proanthocyanidins were likely to inhibit melanogenesis or even melanocyte proliferation only in the chloasma area. The human skin is constantly exposed to UV radiation. UV radiation generates reactive oxygen species (ROS) and leads to oxidative stress. This causes a cascade of erythema and inflammatory reactions, which may be considered as crucial factors affecting the pathogenesis of melasma. Proanthocyanidin has been proven to have a significant free radical scavenging activity in vitro and anti-edema effects in vivo [93]. Additionally, the oral intake of pine bark proanthocyanidins have led to significant reduction in the pigmentation of age spots in photoaged facial skin. The results may be attributed to the antioxidative and anti-inflammatory effects of proanthocyanidins.

The physical, chemical, and biological features of proanthocyanidins depend largely on their structure including the type of flavan-3-ol, particularly on their degree of polymerization. We failed to evaluate the structure–activity relationship (SAR) in each effect because of not so many results of the intervention trial. Proanthocyanidins functioning in terms of absorption and metabolism in human is not yet fully understood. The intestinal cell wall is permeable to proanthocyanidin dimers and trimers, as shown in both in vitro [94] and in vivo experiments [95]. Proanthocyanidins with a degree of polymerisation (DP) below 3 are depolymerised into mixtures of epicatechin monomers and dimers in the acidic environment of the stomach [94] and absorbed by the small intestine [94,95]. However, it is also proposed that a food bolus has a buffering effect, making the acidic conditions milder than that required for proanthocyanidin breakdown. Although proanthocyanidin dimers B1 and B2 can be detected in human plasma [95,96], their absorption is minor, estimated to be more than 100-fold lower than that of the corresponding flavan-3-ol monomers [95]. The cell layer in the intestines is also permeable to oligomeric proanthocyanidins but not polymeric ones [94,95]. Polymeric proanthocyanidins with a DP up to 10 move to the small intestines intact and are mainly degraded by colonic microflora in the cecum and large intestine [97]. Further research is needed to better investigate the active metabolites of proanthocyanidins and detail the mechanism in humans.

## 5. Conclusions

Proanthocyanidins have many effects such as antioxidant, anti-inflammatory, and antimicrobial activities, with a great diversity of complementary alternative therapy supported by sufficient scientific evidence. The consumption of proanthocyanidins can greatly contribute to health promotion in middle-aged and elderly women.

## Figures and Tables

**Figure 1 nutrients-12-03833-f001:**
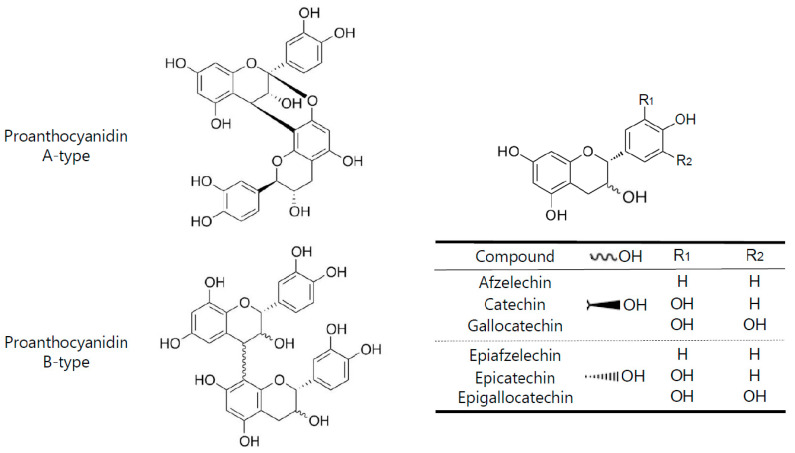
Basic chemical features of A-and B-type proanthocyanidinsand their monomeric units.

**Figure 2 nutrients-12-03833-f002:**
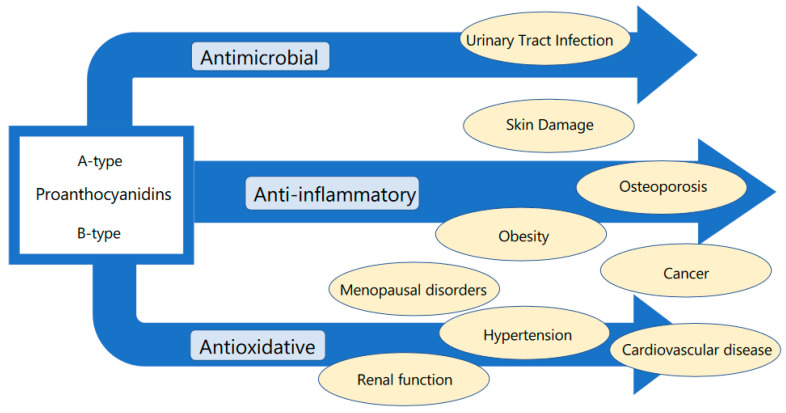
Relationship between the effect of proanthocyanidins and disease field.

**Table 1 nutrients-12-03833-t001:** A summary of current evidence of studies regarding the effects of proanthocyanidins for middle-aged and elderly women.

Studies (Ref. No.)	Study Design	Period	Age	Contents	Main Results
Menopausal Disorders (Including Mental Symptoms)					
Terauchi [17]	RCT	8 weeks	40–60	grape seed extract proanthocyanidins 100 mg/day or 200 mg/day	improvement of HADS-anxiety, HADS-depression and AIS score
Kohama [18]	RCT	12 weeks	42–58	pine bark extract proanthocyanidins 30 mg/day	improvement of vasomotor and insomnia/sleep problem symptoms and Kupperman’s index
Yang [19]	RCT	6 months	45–55	pine bark extract proanthocyanidins 200 mg/day	improvement of climacteric symptoms
Chang [20]	Prospective cohort	10 years	65-	diet-derived proanthocyanidins	lowering of depression risk
Cancer					
Touvier [21]	Prospective cohort	14 years	55-	diet-derived proanthocyanidins	lowering of breast cancer risks
Rossi [22]	case-control	15 years	60–61 (median)	diet-derived proanthocyanidins	lowering of endometrial cancer risk
Culter [23]	Prospective cohort	19 years	55–69	diet-derived proanthocyanidins	decreasing of lung cancer incidence
Hypertension					
Terauchi [17]	RCT	8 weeks	40–60	grape seed extract proanthocyanidins 100 mg/day or 200 mg/day	reducing of SBP and DBP
Odai [24]	RCT	12 weeks	40–64	grape seed extract proanthocyanidins 200 mg/day or 400 mg/day	reducing of SBP and DBP
Lajous [25]	Prospective cohort	16 years	45–58	diet-derived proanthocyanidins	lowering of hypertension rate
Cardiovascular disease					
Odai [24]	RCT	12 weeks	40–64	grape seed extract proanthocyanidins 200 mg/day or 400 mg/day	Improvement of stiffness parameter, distensibility, Eincand PWV
Zhao [26]	RCT	4 weeks	42–53	apple proanthocyanidins	decreasing of oxLDL-b2GPI
Mink [27]	Prospective cohort	13 years	55–69	diet-derived proanthocyanidins	inverse association with coronary heart disease mortality
McCullough [28]	Prospective cohort	7 years	68.9 ± 6.2	diet-derived proanthocyanidins	reduction in cardiovascular disease risk
Jennings [29]	Prospective cohort	12 years	53 (median)	diet-derived proanthocyanidins	lowering of fat mass ratio (FMR)
Obesity					
Tresserra-Rimbau [30]	Prospective cohort	4 years	60–75	diet-derived proanthocyanidins	inverse association with overweight and obese
Kim [31]	Prospective cohort	15 years	45.0 ± 0.2	diet-derived proanthocyanidins	inverse association with abdominal obesity
Osteoporosis					
Panahande [32]	RCT	12 weeks	50–65	pine bark extract proanthocyanidins 250 mg/day	increase of P1NP and BAP levels, decrease of CTx1 levels
Zhang [33]	Prospective cohort	12 weeks	56–63	diet-derived proanthocyanidins	Increase of bone mineral density at the whole body, femoral neck and lumbar spine
Urinary Tract Infection (UTI)					
Takahashi [34]	RCT	6 months	50-	cranberry proanthocyanidins 40 mg/day (cranberry juice)	prevention of the recurrence of UTI
Maki [35]	RCT	24 weeks	40–41 (average)	cranberry proanthocyanidins 41 mg/day (cranberry juice)	lowered the number of clinical UTI episodes
Vostalova [36]	RCT	6 months	18–75	cranberry powder 500 mg/day (proanthocyanidins 0.56%)	reduction of the risk of symptomatic UTI
Renal Function					
Ivey [37]	Prospective cohort	5 years	80 ± 3	diet-derived proanthocyanidins	improvement of renal function and reduction of risk of chronic kidney disease and renal disease associated events
Skin Damage					
Yamakoshi [38]	single-armed	12 months	34–58	grape seed extract proanthocyanidins 160 mg/d	improvement of chloasma and decreasing melanin index
Evangeline [39]	RCT	8 weeks	18–60	cranberry proanthocyanidins 24 mg/day	decreasing the degree of pigmentation in the malar regions and improvement of the melasma area and severity index
Furumura [40]	RCT	12 weeks	31–59	pine bark extract proanthocyanidins 100 mg/day	improvement of scores for solar lentigines, mottled pigmentation, roughness, wrinkles, and swelling

HADS: Hospital Anxiety and Depression Scale; SBP: systolic blood pressure; DBP: diastolic blood pressure; BAP: bone alkaline phosphatase; P1NP: procollagen type 1 amino-terminal propeptide; CTx1: C-terminal telopeptide of type I collagen.

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
