# Peer review of "The Diverse Efficacy of Food-Derived Proanthocyanidins for Middle-Aged and Elderly Women"

_nutrients, 2020, doi:10.3390/nu12123833_

Round 1
Reviewer 1 Report
In the review “The Diverse Efficacy of Food-Derived Proanthocyanidins for Middle-Aged and Elderly Women” the authors have conducted an incredibly detailed overview of the effect of the assumption of these molecules on the health of Middle-Aged and Elderly Women. The topic of this review is interesting and well-focused.
The review is well constructed, and the sections are well separated.
I have only a few suggestions:
1-As authors describe the type of linkages in proanthocyanidins, I suggest inserting a figure reporting the general structures of A-type and B-type proanthocyanidins and the examples of (epi)catechin, (epi)afzelechin and (epi)gallocatechin.
2-I suggest inserting a scheme or a picture reporting the studies on the effects of food-derived proanthocyanidins assumption on women’s health: menopausal disorder, cancer, hypertension, cardiovascular diseases, etc.
Author Response
Dear Reviewer 1,
Thank you for your suggestions.
At first, I made the figure of chemimcal features of A- and B- type proanthocyanidins and their monomeric utits, as Figure 1.
Next, according to your second suggestion, I made the figure of the scheme for the relationship between the effect of proanthocyanidins and disease filed, as Figure 2.
Please see the attachment of the PDF files.
Best regards,
Toru Izumi

Reviewer 2 Report
The review aims to highlight aspects related to the effects of food-derived proanthocyanidins for middle-aged and elderly women, focusing on menopausal disorders, cancer, hypertension, CVD, obesity, osteoporosis, UTI, renal function, and skin damage.
The review summarizes well all conditions to which proanthocyanidins have been studied in middle-aged and elderly women and the discussion is well-developed; however, there are some major aspects that should be clarified to accept the manuscript for its publication.
It is not clear what methodology was used by authors to collect the references cited in the paper. Although it is not mandatory, it is highly recommended to adhere to a well-established methodology for a systematic review, like PRISMA for instance. This will help to clarify the reader the reasons for inclusion and exclusion of papers. A table summarizing the designs, primary and secondary endpoints, and results of each study will benefit the paper and will also reduce the length in the reporting of results.
An important point for consideration by authors, and that will not diminish the value of the paper, is to mention the endpoints that have not been affected by proanthocyanidins. As it is written now, the paper only reports positive effects, and not diverse effects.
There are also some minor observations on points that might have skip author’s attention. For example, some references should be properly cited (e.g. lines 31, 371) or typos (e.g. lines 132, 197, 307, 308)
Author Response
Dear Reviewer 2,
Thank you for your suggestions.
At first, I added new section, "Search Strategy and Study Selection" as the methodology to collect the refferences in revised manuscript.
Secondly, I made the table reporting the summary of the evidence of the studies, as Table 1.
At last, although I researched the effects that have not been affected by proanthocyanidins, I could not find the corresponding reports of the effects for middle-aged and elderly women. I added the above sentense in "3. Results" section.
Please see the attachment of the PDF files.
Best regards,
Toru Izumi